

# Paranoia and conspiracy: group cohesion increases harmful intent attribution in the Trust Game

Anna Greenburgh[1], Vaughan Bell[2,3] and Nichola Raihani[1]

[1] Department of Experimental Psychology, University College London, London, United Kingdom
[2] Department of Clinical, Education and Health Psychology, University College London, London, United Kingdom
[3] Psychological Interventions Clinic for Outpatients with Psychosis (PICuP), South London & Maudsley NHS Foundation Trust, London, United Kingdon

## ABSTRACT

Current theories argue that hyper-sensitisation of social threat perception is central to paranoia. Affected people often also report misperceptions of group cohesion (conspiracy) but little is known about the cognitive mechanisms underpinning this conspiracy thinking in live interactions. In a pre-registered experimental study, we used a large-scale game theory approach ($N > 1,000$) to test whether the social cohesion of an opposing group affects paranoid attributions in a mixed online and lab-based sample. Participants spanning the full population distribution of paranoia played as proposers in a modified Trust Game: they were allocated a bonus and chose how much money to send to a pair of responders which was quadrupled before reaching these responders. Responders decided how much to return to the proposers through the same process. Participants played in one of two conditions: against a cohesive group who communicated and arrived at a joint decision, or a non-cohesive group who made independent decisions. After the exchange, proposers rated the extent to which the responders' decisions were driven by (i) self-interest and (ii) intent to harm. Although the true motives are ambiguous, cohesive responders were reliably rated by participants as being more strongly motivated by intent to harm, indicating that group cohesion affects social threat perception. Highly paranoid participants attributed harmful intent more strongly overall but were equally reactive to social cohesion as other participants. This suggests that paranoia involves a generally lowered threshold for social threat detection but with an intact sensitivity for cohesion-related group characteristics.

# INTRODUCTION

Although paranoia is the most common positive symptom of psychosis (*Freeman, 2007*), it is also present to varying degrees in the general population. Paranoia can range from mildly exaggerated concerns about how others see us, to frank and disabling paranoid delusions of conspiratorial harm (*Freeman & Garety, 2014*; *Taylor, Freeman & Ronald, 2016*; *Elahi et al., 2017*). Estimations suggest a third of the general population frequently experience paranoid

Corresponding author
Nichola Raihani, n.raihani@ucl.ac.uk, nicholaraihani@gmail.com

thoughts (*Freeman et al., 2005*). Paranoid ideation follows an exponential distribution where increasingly severe thoughts are increasingly rare (*Bebbington et al., 2013*).

One well-established component of paranoia is an alteration to social threat perception. Patients with persecutory delusions have better memory for threat-related words (*Bentall, Kaney & Bowen-jones, 1995*) and actively paranoid patients diagnosed with schizophrenia over-perceive anger in neutral faces (*Pinkham et al., 2011*). Highly paranoid members of the general population over-estimate chances of future victimisation (*Jack & Egan, 2016*) and are more likely to perceive harmful intent in ambiguous social exchanges (*Raihani & Bell, 2017a*; *Raihani & Bell, 2017b*).

Paranoia also frequently involves problems with accurately judging the intentions of groups, rather than individuals (*Raihani & Bell, 2018*). In clinical studies, concerns about conspiracy are a well-established component of paranoia that have been documented from early in the history of psychiatry (*Harper, 1994*) and form part of current definitions (*Oyebode, 2008*). Here, conspiracy perception refers to concerns about being persecuted by a group of others who are coordinated in attempts to harm the individual, but who do not correspond to any collection of people with these aims—something *Cameron (1959)* conceptualised as the 'paranoid pseudocommunity'. This is distinct from belief in public conspiracy theories more broadly, which describe conspiratorial explanations of important historical events that are not centred on the believer (*Douglas, Sutton & Cichocka, 2017*). Indeed, recent cross-cultural research highlights that that while paranoia and conspiracy thinking (belief in public conspiracy theories) are associated, they are divergent constructs (*Imhoff & Lamberty, 2018*).

Rather than viewing paranoia only as a symptom of a mental disorder, recent research has suggested that it might be more accurate to view paranoia across the full spectrum of severity as part of a normally-functioning human psychology that evolved in the context of detecting and avoiding social threat (*Green & Phillips, 2004*; *Raihani & Bell, 2017a*; *Raihani & Bell, 2017b*). Studies that have examined social threat processing as a broader psychological mechanism suggest it is likely to be supported by a specialised, evolved mechanism that is attuned to threats from both individuals and groups. The Coalitional Index Model (*Boyer, Firat & van Leeuwen, 2015*) proposes that individuals can judge cues of social threat and support and can integrate them to encode their own vulnerability in any given environment. Under this hypothesis, activation of the 'coalitional safety index' triggers a set of physiological, cognitive and behavioural responses attuned to the perceived degree of social threat. Threat inputs to this mechanism include coalitional identity, as supported by evidence that out-group cues increase anxiety and fear responses (e.g., *Navarrete et al., 2009*; *Hart et al., 2000*).

Such a psychological mechanism to detect social threat should be sensitive to the cohesiveness of rival groups. More cohesive groups are more able to act towards a common goal. Therefore, cohesive opponent groups should be perceived as more threatening than similar-sized but non-cohesive groups (*Boyer, Firat & van Leeuwen, 2015*). Indeed, group entitativity (its perceived unity) increases negative cognitive and behavioural responses towards the group (*Campbell, 1958*; *Hamilton & Sherman, 1996*). High-entitativity groups are also perceived as more morally suspicious (*Newheiser, Sawaoka & Dovidio, 2012*),

conspiratorial (*Grzesiak-Feldman & Suszek, 2008*), more capable of negative retaliation (*Dasgupta, Banaji & Abelson, 1999*) and evoke increased negative stereotyping (*Spencer-Rodgers, Hamilton & Sherman, 2007*).

One clear implication of these studies is that individuals will exhibit increased threat responses when in the presence of cohesive compared to non-cohesive groups. However, despite the clear overlap with paranoia both conceptually and clinically, the effects of group cohesiveness on threat perception have never been tested in this domain. Research is needed in this area to further our understanding of social experience of paranoia, and how mechanisms of social cognition function in paranoia.

Various hypotheses regarding paranoia and group cohesion emerge. The first is that paranoia may involve an exaggerated reaction to group cohesion (a social over-sensitivity bias) so that highly paranoid people will increase their attributions of harmful intent to cohesive compared to non-cohesive groups, over and above the degree to which low-paranoia individuals do. Another possible pattern is that paranoia may involve a misperception of group cohesiveness such that all groups are perceived to be cohesive. In this case highly paranoid people will attribute equally-high levels of harmful intent to cohesive and non-cohesive groups alike, unlike low-paranoia individuals who will attribute greater levels of harmful intent to cohesive compared to non-cohesive groups. Finally, it could be that paranoia simply involves a higher baseline of harmful intent attribution—a general tendency to over-attribute harmful intent in social situations when compared to low-paranoia individuals—but show the same pattern of reactivity to group cohesiveness as low-paranoia individuals.

Here, we report a pre-registered study testing these predictions. We recruited a dual online and offline sample, including a large online sample covering the population distribution of paranoid ideation, and a lab-based panels of participants who formed cohesive and non-cohesive pairs to respond to decisions made by the online participants. Previous studies have demonstrated that large-scale game theory approaches are effective in capturing live paranoid attributions and testing how paranoid attributions are modified by experimentally-induced social threat (*Raihani & Bell, 2017a*; *Raihani & Bell, 2017b*; *Saalfeld et al., 2018*). For example, modified Dictator Games have revealed that individuals spanning the full paranoia spectrum rate unfair Dictators as intending more harm than fair Dictators, signifying that fairness is used as a threat cue irrespective of an individual's level of paranoid ideation (*Raihani & Bell, 2017a*; *Raihani & Bell, 2017b*). Similarly, interacting with higher status or out-group partners triggers exaggerated attributions of harmful intent in otherwise ambiguous interactions (*Saalfeld et al., 2018*).

In the current study, participants played as proposers in an adapted Trust Game in one of two conditions, where they interacted (i) against a pair of cohesive responders or (ii) against a same-sized group of non-cohesive responders. Participants were given an initial sum of money ($0.50) and were asked to indicate what proportion they would like to send ($0.25 or $0.05) to a pair of responders. Participants kept any of the money they did not send to the responders. Participants were told that any amount they sent would be quadrupled by the experimenter, and that the responders could decide whether to be fair and return half of the increased amount to the participant or to be unfair and return

nothing to the participant. Participants were randomly allocated to either fair or unfair responder pairs.

The cohesiveness of the responder pair was characterised by whether responders could communicate with one another or not. In the cohesive condition, responders had to agree on their decision, whereas in the non-cohesive condition responders came to their decision independently.

Participants chose how much to send to the responders and were then presented with the responders' decision. After finding out how much the responders returned to them, participants rated the extent to which they believed the responders' decisions to be driven by (i) self-interest and (ii) harmful intent, using two separate slider scales. The true motives underpinning responder decisions in this game are ambiguous and could plausibly reflect either self-interest (desire to keep more money) or harmful intent (desire to exploit the participant). Previous work indicates that paranoia positively predicts harmful intent attributions but not self-interest attributions in similarly ambiguous settings (*Raihani & Bell, 2017a*; *Raihani & Bell, 2017b*; *Saalfeld et al., 2018*).

## MATERIALS AND METHODS

This study was approved by the UCL Ethics board, under project 3720/001. Participation was voluntary and informed consent was obtained from all participants prior to their participation.

### Participants

The behavioural data were collected in August 2017; the GPTS data were collected in December 2016. We recruited 1,172 participants to the study (682 female). 1,164 of these participants were recruited via Amazon Mechanical Turk (MTurk; http://www.mturk.com), whose decisions were the focus of the analysis. Participants for this study were recruited from an existing database of ∼3,500 participants for whom we already had data on pre-existing paranoia. This data was collected in December 2016 for a previous study published in June 2017 (*Raihani & Bell, 2017a*; *Raihani & Bell, 2017b*). The stability of paranoia scores over time was confirmed by re-collecting the GPTS scores of 420 participants from December 2016 in August 2017 ($r_s = 0.68$, $p < .001$).

We aimed to recruit a minimum of 2,000 participants from this original database for this study but we pre-registered a stopping rule that indicated we would stop data collection after responses were slower than 20 per day. In addition, 8 participants took part in the study in the laboratory and acted as paired responders to the online participants and were configured as cohesive (communicated and made decisions together) or non-cohesive (did not communicate and made decisions separately) pairs in deciding the response. All MTurk participants were based in the US. The mean age of the (online) participants was 38.2 ($SD = 12.3$).

### Design

We used a between-subject design, where the independent variables were the cohesiveness of the responder group in the Trust Game (non-cohesive / cohesive), fairness of responder

decision (fair/unfair) and paranoia (as measured by the GPTS). Dependent variables were attributions of (i) harmful intent and (ii) self-interest made by proposers to the responder group. Predictions were pre-registered at https://aspredicted.org/see_one.php?a_id=4809.

## Materials and Procedure

We used the *Green et al.*'s (*2008*) Paranoid Thought Scales (GPTS) to measure pre-existing paranoid ideation. The GPTS is a reliable and valid scale for measuring paranoid ideation across the full clinical and non-clinical spectrum. It is a 32-item scale, consisting of two 16-item subscales that measure feelings of social reference and persecution, respectively. Participants were asked to endorse each item on a five-point Likert scale from 'Not at all' to 'Totally'. Responses are summed to provide final scores, which can range from 32–160, where higher scores indicate a greater degree of paranoia. This variable will be referred to as 'pre-existing paranoia'.

We used a modified Trust Game as the main behavioural paradigm. A standard Trust Game (*Berg, Dickhaut & McCabe, 1995*) involves two participants, a proposer and a responder. The proposer is allocated a sum of money and must decide how much to send to the responder. Any amount they send is multiplied by a given factor, and the responder must decide how much of the resultant amount to send back to the proposer. Our paradigm modified this design in two main fashions: two responders played against a single proposer in each game, and we systemically manipulated the cohesiveness of the responder pair.

For the current study, proposers were recruited online and responders were lab-based participants. On the online platform, proposers were allocated an initial sum of money ($0.50) and were asked to indicate what proportion of this they would like to send ($0.25 or $0.05) to the pair of responders, where the remainder would be theirs to keep. Proposers were told that any amount they sent to responders would be quadrupled. Responders decided how much of that amount to return to the proposer. Responder decisions were pre-collected (proposers were aware of this), and ex-post matching was used to assign proposers to responder pairs (*Raihani, Mace & Lamba, 2013*). Responders were shown all possible decisions of proposers and were asked to decide how much money they would return given each proposer decision.

Cohesiveness of the responder pair was characterised by whether they had to communicate and make their decision as a pair. In the cohesive condition, responders communicated through an online chat function and were required to come to a joint decision regarding much money to return to the proposer. In the non-cohesive condition, responders did not communicate: they came to their decision independently and they were subsequently matched with other responders who made the same decision. The cohesiveness of responders was clearly communicated to the proposers. This operationalization of cohesiveness was taken from key characteristics discussed in the group cohesiveness and entitativity literature. This literature outlines many cues of group cohesion, including group member interaction, agreement, and common goals (*Lott & Lott, 1961*; *Campbell, 1958*; *Lickel, Hamilton & Sherman, 2001*). The online, anonymous nature of our design meant that we were unable to use many cues that are associated with group cohesion,

such as dress and distinct language (*Boyer, Firat & van Leeuwen, 2015*). Despite using a subset of possible indicators of group cohesiveness, we confirmed that our manipulation impacted perception of cohesiveness through a pre-registered manipulation check, which showed that 486 / 587 (83%) participants in the cohesive condition correctly perceived the responder pair as cohesive. For the non-cohesive condition, 504 /564 (89%) participants correctly perceived the responder pair as non-cohesive. Binomial tests confirmed that participants reliably perceived the condition to which they were assigned ($p < .001$). Furthermore, correct perception of group cohesiveness did not vary with paranoia.

After reading the instructions, participants were asked to answer three comprehension questions to confirm they had understood the game (see Supplemental Information). We conducted pre-registered analyses to determine any effects of comprehension on the results (see below).

After making their decisions, proposers were informed of the responders' decision (fair/unfair). They were asked to complete two ratings (using slider bars initialised at 50) on a scale of 1 to 100 to what extent they believed the responders' decision was motivated by (i) a desire to earn more, and (ii) a desire to reduce the proposer's bonus. These were the measures of self-interest and harmful intent attributions, respectively.

After completing these ratings, proposers were asked to answer a manipulation check, with three possible answers, to measure how cohesive they perceived the responders to be: 1) responders made their decisions as a team, 2) responders made their decisions separately or 3) unsure.

We pre-registered six main predictions of the study before collecting behavioural data.

(P1)     Proposers will attribute increased harmful intent to unfair compared to fair responders.

(P2)     Proposers will attribute increased self-interest to unfair compared to fair responders.

(P3)     Proposers will attribute increased harmful intent to responders in cohesive compared to non-cohesive groups.

(P4)     More paranoid proposers will make higher harmful intent attributions overall.

(P5)     More paranoid proposers will show dysregulated responses to opponent cohesion (there will be a paranoia x cohesion interaction on attributions of harmful intent)

(P6)     Attributions of self-interest will not be related to paranoia.

### Statistical approach

We employed an information-theoretic (IT) approach with multi-model averaging (*Burnham & Anderson, 2002*; *Grueber et al., 2011*) to compare the explanatory power of different input variables. The IT approach has many advantages and is widely used in ecology research (see *Whittingham et al., 2006* for a detailed review). Unlike null hypothesis significance testing, our method does not employ arbitrary $p$ values as indicators of significance. Rather, we examine the relative support for each model as given by AICc (Akaike Information Criterion, corrected for small sample sizes) values. AIC values

estimate the goodness-of-fit of the model to the data (with lower AIC values indicating greater support), while penalising models for the inclusion of additional explanatory terms. Selecting models based on AIC values therefore prioritises parsimony: models should be as simple as possible, but no simpler. This philosophy represents a balance between over-fitting and under-fitting models with too many or too few parameters, respectively.

Our analysis procedure proceeded in four key steps (following *Grueber et al., 2011*): (i) we specified a full global model containing all terms of interest; (ii) we compared all possible subsets of this model, containing all possible combinations of terms, to one another; (iii) we obtained a 'top model set', containing the subset of models which had equal support (AICc values within two units of one another) and (iv) we averaged across this top model set to derive estimates and confidence intervals for the explanatory terms contained in the top model set. Multi-model averaging therefore reduces the reliance on a single 'best' model to obtain parameter estimates and confidence intervals and instead takes into account the fact that there is uncertainty over the true 'best' model.

We conducted two broad analyses with (i) harmful intent and (ii) self-interest as the respective response terms. We present full (unconditional) model averaged estimates and confidence intervals here, which are more conservative estimates. Due to the extreme skew of the response variables, we converted each one into a 5-level ordered categorical variable (*Raihani & Bell, 2017a*; *Raihani & Bell, 2017b*). Each term was specified as an ordinal categorical response term in an ordinal logistic regression model, using the *clm* function in the ordinal package (*Christensen, 2015*). In each model, we included the following explanatory terms: paranoia, trust decision (large/small investment), condition (cohesive/non-cohesive), fairness (fair/ unfair), gender and age. Binary input variables were recoded as dummy (1/0) variables.

All continuous input variables were standardized (*Gelman, 2008*) and binary input variables were centered, so estimates can be considered on the same scale. We included incomprehension (whether subjects answered 1+ comprehension questions incorrectly) in the analyses as a binary dummy variable to control for the effect of failed comprehension on responses. Where comprehension was found to have a meaningful effect, we re-ran analyses including and excluding non-comprehenders to determine whether the models qualitatively changed (no qualitative differences in the results presented below were found).

### Data and code availability
The data and code to reproduce the analyses in this study are available in OSF with the identifier DOI 10.17605/OSF.IO/W7YQX.

## RESULTS
The mean ($\pm$SEM) paranoia score was $50.7 \pm 0.67$, reflecting the combined scores of social reference ($27.1 \pm 0.35$) and persecution ($23.5 \pm 0.36$) subscales, respectively. Paranoia scores ranged from 32 to 158, with 54 participants (4.63%) being over the clinical mean of 101.9 reported in *Green et al. (2008)*.

As predicted, participants attributed more harmful intent to cohesive compared to non-cohesive opponents (estimate: 0.39, CI [0.15–0.64]; Fig. 1, Table 1) but paranoia
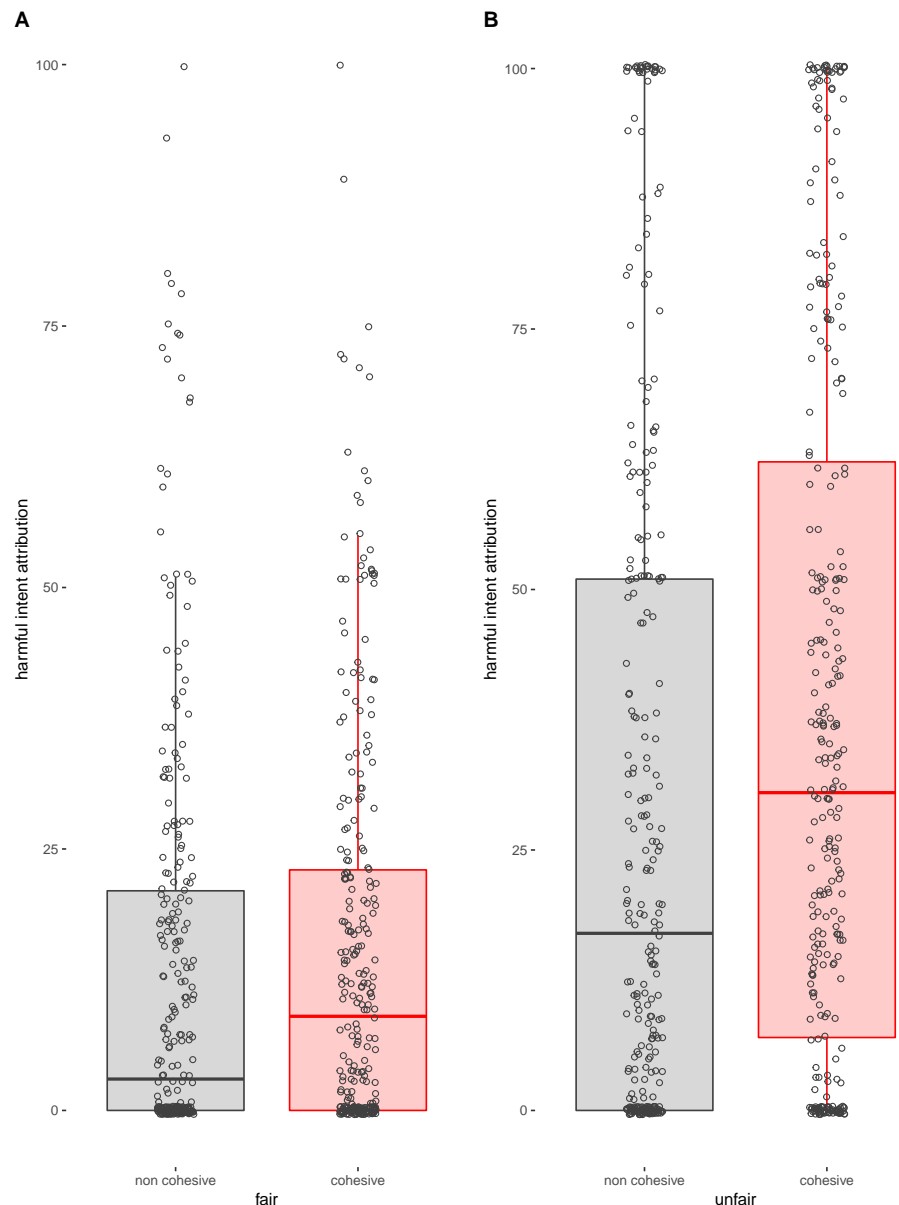

**Figure 1  Box plot to show harmful intent attribution made by proposers concerning pairs of responders.** Each data point indicates the harmful intent each proposer attributed to the responder team they interacted with, according to the cohesiveness of responders, and whether responders' decisions were fair (A) or unfair (B).

did not interact with opponent cohesiveness to exaggerate this effect (estimate: $-0.06$, CI $[-0.35-0.24]$, Table 1). All predictions regarding the validity of the paradigm were supported. Participants attributed more harmful intent to unfair rather than fair opponents (estimate: $-1.19$, CI $[-1.43--0.95]$, Table 1) and more self-interest to unfair rather than fair opponents (estimate: $-3.37$, CI $[-3.67--3.06]$, Table 2). In addition, paranoia independently and positively predicted harmful intent attribution (estimate: 0.77, CI

**Table 1** **Information for the ordered logistic regression investigating the attribution of harmful intent to responder pairs in the trust game.** Model average estimates, unconditional standard errors, confidence intervals and relative importance for the terms included in the top model set for the ordered logistic regression investigating the attribution of harmful intent to responder pairs in the trust game. See Supplemental Information for top model set. Reference levels are shown in parentheses.

| Parameter | Estimate | Unconditional SE | Confidence interval | Relative importance |
|---|---|---|---|---|
| *Intercept 1 \|2* | *0.42* | *0.07* | *(0.29, 0.54)* | |
| *Intercept 2 \|3* | *1.31* | *0.08* | *(1.17, 1.46)* | |
| *Intercept 3 \|4* | *2.04* | *0.09* | *(1.86, 2.22)* | |
| *Intercept 4 \|5* | *2.76* | *0.12* | *(2.53, 2.99)* | |
| Cohesion (1 = cohesive) | 0.39 | 0.12 | (0.15, 0.63) | 1.00 |
| Fairness (1 = fair) | −1.19 | 0.12 | (−1.44, −0.95) | 1.00 |
| Gender (1 = male) | −0.36 | 0.13 | (−0.60, −0.11) | 1.00 |
| Comprehension (1 = >1 comprehension failure) | 0.86 | 0.22 | (0.43, 1.29) | 1.00 |
| Trust decision (1 = sent larger amount) | −0.34 | 0.14 | (−0.62, −0.05) | 1.00 |
| Paranoia | 0.77 | 0.11 | (0.55, 1.00) | 1.00 |
| Cohesion:Paranoia | −0.06 | 0.15 | (−0.35, 0.24) | 0.32 |

**Table 2** **Information for the ordered logistic regression investigating the attribution of self-interest to responder pairs in the trust game.** Model averaged estimates, unconditional standard errors, confidence intervals and relative importance for the terms included in the top model set for the ordered logistic regression investigating the attribution of self-interest to responder pairs in the trust game. See Supplemental Information for top model set. Reference levels are shown in parentheses.

| Parameter | Estimate | Unconditional SE | Confidence Interval | Relative importance |
|---|---|---|---|---|
| *Intercept 1 \|2* | *−2.84* | *0.11* | *(−3.06, −2.63)* | |
| *Intercept 2 \|3* | *−1.97* | *0.09* | *(−2.15, −1.78)* | |
| *Intercept 3 \|4* | *−1.26* | *0.09* | *(−1.43, −1.09)* | |
| *Intercept 4 \|5* | *0.05* | *0.08* | *(−0.10, 0.21)* | |
| Fairness (1 =fair) | −3.37 | 0.15 | (−3.67, −3.06) | 1.00 |
| Gender (1 = male) | −0.07 | 0.12 | (−0.31, 0.16) | 0.43 |
| Comprehension (1 = >1 comprehension failure) | −0.09 | 0.19 | (−0.46, 0.29) | 0.34 |
| Trust decision (1 = sent higher amount) | 0.06 | 0.12 | (−0.18, 0.29) | 0.35 |
| Cohesion | −0.09 | 0.03 | (−0.06, 0.06) | 0.06 |
| Paranoia | −0.0021 | 0.03 | (−0.07, 0.06) | 0.06 |

[0.55–1.00]; Table 1, Fig. 2) and there was no effect of paranoia (Fig. 2) or opponent cohesiveness on attributions of self-interest (Table 2, Fig. 2).

## DISCUSSION

We investigated the effects of pre-existing paranoia on harmful intent attribution when interacting with cohesive and non-cohesive groups. Participants played as proposers in

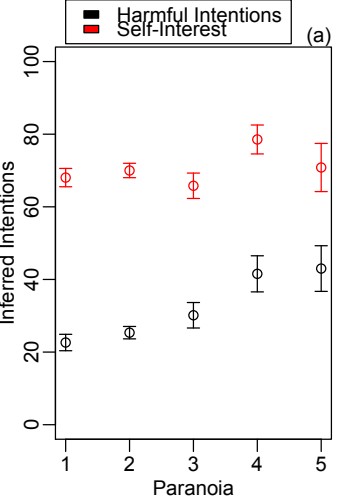
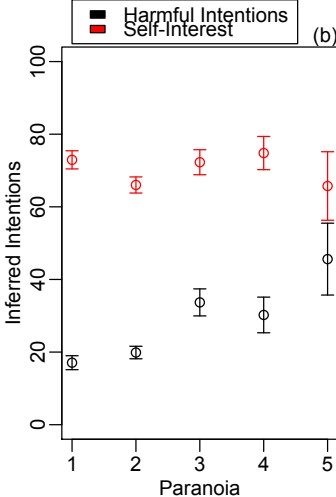

**Figure 2  Mean harmful intent and self-interest attributions made by participants.** Data points indicate the harmful intent (black) and self-interest (red) attributions made by participants in (A) the cohesive responder condition and (B) the non-cohesive responder condition. Means and standard errors are generated from raw data. Paranoia was converted to a five-level categorical variable for ease of visualisation, although it was included as a continuous term in the models.

an adapted Trust Game against a pair of cohesive or non-cohesive responders. Although the intentions of responders are ambiguous in this task, cohesive responder groups were perceived as more malevolent than non-cohesive responder groups, as measured by harmful intent attributions made by proposers. Paranoia was associated with a general tendency to make stronger harmful intent attributions in all situations but highly paranoid individuals showed the same increase in harmful intent attribution when interacting with cohesive groups as individuals with lower paranoia. This suggests that pre-existing paranoia reflects a generally heightened tendency for threat perception but an intact sensitivity to group cohesion. Therefore, there was no evidence of a qualitatively nor quantitatively different response to the social context that may reflect a dysregulation in perceiving group interactions.

This finding converges with studies that found no interaction between paranoia or psychosis and social threat on behavioural and physiological responses (*Veling et al., 2016*; *Counotte et al., 2017*; *Saalfeld et al., 2018*). Together, these studies suggest that paranoia is associated with a lower threshold for detecting social threat; but the scaling of response to changes in social threat is similar in all individuals, irrespective of where they lie on the paranoia spectrum. This goes against theories that suggest paranoia involves an increased reactivity to social threat (*Yiend et al., 2017*; *Savulich et al., 2015*; *Pot-Kolder et al., 2017*); and against a generalised insensitivity to social information (*Fett et al., 2012*; *Fett et al., 2016*; *Gromann et al., 2013*).

Our results support the hypothesis that group cohesiveness acts as a social threat cue, since paranoid attributions increased in response to greater group cohesiveness. This coincides with evidence that tightly-formed groups are seen as more legitimate subject of

conspiracy theories (*Grzesiak-Feldman & Suszek, 2008*). In light of the Coalitional Index Model (*Boyer, Firat & van Leeuwen, 2015*), this would imply that group cohesiveness is an input into the Coalitional Safety Index. Changes in cohesiveness of the responder group affected proposers' attributions of harmful intent but not self-interest: the responder group was perceived as no more self-interested in conditions of high compared to low cohesiveness, even though cohesive responders were perceived as more malevolent than non-cohesive responders. This can be viewed as a conceptual replication of data showing other social threat cues, group affiliation and social status, independently alter attributions of harmful intent attribution, but not self-interest (*Saalfeld et al., 2018*).

Despite capturing paranoid ideation scores common to patients with psychosis, it is unclear to what extent these results would characterise patients with paranoid delusions. The mean paranoia score in our sample was similar to that of the non-clinical sample in *Green et al.*'s (*2008*) original study, with a similar percentage of participants scoring above the clinical mean. However, we did not record whether any participant had a clinical diagnosis and assessing the presence of delusions is best completed with a structured clinical interview, rather than self-report measures. It would be surprising if in such a large sample no participants with delusions were present but we have also noted that severe presentations of delusional paranoia are likely to involve specific changes to social perception that are not present in non-delusional presentations.

The online nature of the study permitted a large sample size, as well as a more representative sample than typical samples from typical university undergraduate and community participant pools (*Berinsky, Huber & Lenz, 2012*). However, MTurk participants still show important demographic differences to the general population (*Huff & Tingley, 2015*) and in terms of mental health show higher levels of social anxiety although they are not more likely to report clinically relevant emotional dysregulation than the wider population (*Shapiro, Chandler & Mueller, 2013*). Furthermore, given all our participants were based in the US it is not clear how well these effects might generalise more widely.

## CONCLUSIONS

Our results replicate and extend existing research in paranoia. Firstly, they suggest that perceptions of opponent group cohesion moderate live paranoid ideation, where more cohesive groups evoke greater perception of threat. Secondly, these results support evidence that those scoring high in paranoia in the general population have a greater tendency to perceive conspiracy, but these attributions of malevolent intent show intact scaling in response to changes in social threat.

### Funding

Anna Greenburgh is funded by the Royal Society. Vaughan Bell is supported by a Wellcome Trust Seed Award in Science [200589/Z/16/Z]. Nichola Raihani is supported by a Royal

Society University Research Fellowship and the Leverhulme Trust The funders had no role in study design, data collection and analysis, decision to publish, or preparation of the manuscript.

## Grant Disclosures

The following grant information was disclosed by the authors:
Royal Society.
Wellcome Trust Seed Award in Science: 200589/Z/16/Z.
Royal Society University Research Fellowship.

## Competing Interests

The authors declare there are no competing interests. Vaughan Bell is employed by the South London and Maudsley NHS Foundation Trust.

## Author Contributions

- Anna Greenburgh and Nichola Raihani conceived and designed the experiments, performed the experiments, analyzed the data, contributed reagents/materials/analysis tools, prepared figures and/or tables, authored or reviewed drafts of the paper, approved the final draft.
- Vaughan Bell conceived and designed the experiments, contributed reagents/materials/-analysis tools, authored or reviewed drafts of the paper, approved the final draft.

## Human Ethics

The following information was supplied relating to ethical approvals (i.e., approving body and any reference numbers):
    This study was approved by the UCL Ethics board, under project 3720/001.

## Data Availability

    The data that support the findings of this study and code supporting the statistical analysis used is available at Greenburgh, A., Bell, V., & Raihani, N. (2018, November 6). Conspiracy thinking increases in interactions with cohesive opponents. https://doi.org/10.17605/OSF.IO/W7YQX.

## Supplemental Information

Supplemental information for this article can be found online at http://dx.doi.org/10.7717/peerj.7403#supplemental-information.

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
