# Peer review of "Paranoia and conspiracy: group cohesion increases harmful intent attribution in the Trust Game"

_PeerJ, doi:10.7717/peerj.7403_

## Round 0.1 · original submission · Minor Revisions

Your manuscript has now been seen by 3 reviewers. You will see from their comments below that they find your work of interest, and some constructive points are worth considering. We therefore invite you to revise and resubmit your manuscript, taking into account the points raised. Please highlight all changes in the manuscript text file.

·

Basic reporting

This is a very well written and interesting article. The language was clear and the literature cited was adequate. The authors share the raw data. I have only noticed two minor spelling mistakes:
-line 239: "re-run analyses" should be "re-ran analyses"
-line 314: "shows" should be "show"

Also, the panels in Figure 2 are not labeled. They should be labeled with respect to the caption.

Experimental design

This article aims to test the relationship between paranoia and threat perceptions and whether or not this relationship was moderated by perceptions of group cohesiveness. While they found evidence that paranoia increased sensitivity to threats, their results did not support the prediction that paranoia was related to perceptions of group cohesiveness. Overall the research question was well defined and meaningful. I have only a few comments/suggestions to clarify the writing and experimental design:

1- Somewhere in the introduction it would be good to provide some statistics on how common paranoia is in the general population and what the distribution looks like.
2- In section 2.1, under Participants, the authors indicate that they had data on pre-existing paranoia. They could explain this a little more. When was this data collected, for what purpose? Was the GPTS instrument given again to the participants or is this from this previous data?
3- Under section 2.2.Design, they introduce a new concept: fairness of responder decision. There is not mention of this concept in the theory up until now. This is a bit confusing. I believe the authors should make the theoretical connections between fairness and paranoia and threat more explicit early on.
4- The justification for the Information-theoretic approach is not clear. Why is this method used? What is the main advantage? And how exactly does this method address issues related to multiple hypothesis testing?
5- On page 229, the authors mention they have included an incomprehension variable but it is not clear how this variable was measured. It needs to be specified.

Validity of the findings

I believe the analyses and the overall results are clear and robust. I had one main question and perhaps a suggestion. I wondered if the authors had enough variation on their paranoia variable and whether or not the mean score was very low. Perhaps they could speak to this in the paper. If the think there is enough variation and there are enough people with high scores, perhaps they could do some additional analyses focusing only on the lowest and the highest paranoia people to see if it makes a difference in terms of perceptions of group cohesion.

Reviewer 2 ·

Basic reporting

No comment

Experimental design

No comment

Validity of the findings

No comment

Additional comments

Review “Paranoia and conspiracy: group cohesion increases harmful intent attribution in the Trust Game”

The manuscript is well-written and reports a well-executed experiment. I have several comments about aspects that I found unclear or limitations that might be addressed.

1. Lines 118-122: Cohesiveness of the responder pair was operationalized as whether the responder pair did or did not communicate. Please discuss whether this is a valid operationalization. What are the strengths of this operationalization? What are the limitations?

2. Lines 172-181: Responder decisions were pre-collected in the lab and MTurk participants were aware of this. Please discuss whether this has threatened the internal validity. This kind of online game is very different from regular social interaction in which one party does something and the other party responds. Now the participants did something and then were linked to a response that had already happened in the past. (Right?) Does that matter? I mean, is it possible that highly paranoid participants perceived this situation differently than regular participants, which influenced the behavior or the highly paranoid participants in this task? In other words, please discuss whether the experimental situation may have been perceived differently by highly paranoid participants and whether this has introduced confounds. (I mean, a trust game is a strange game, even when played face-to-face (e.g., why does the experimenter multiply the transfers?), I imagine that someone who is slightly paranoid, could have multiple things to be paranoid about in such a game.)

3. Line 258: Tables 1 and 2 report results from an ordered logistic model. I do not understand this. The outcome variables (attribution of harmful intent, attribution of self-interest) were measured on a continuous scale from 1 to 100 (lines 195-198). So why not do a standard OLS regression to test the hypotheses? I appreciate that the authors did effort to report a sophisticated analysis, but in my view, the analysis should not be more complicated than is needed. (Related to this, I do not fully understand the reported regression coefficients that were averaged over multiple models. This is a bit unusual, so just for communicating the results to readers who may be a bit skeptical toward unfamiliar things, please consider reporting the regular non-averaged regression coefficients.)

4. Lines 279-280: Why would we expect there to be a QUALITATIVELY different response? Does P5 not suggest that the strongly paranoid individuals are more sensitive to group cohesion (e.g., at the slightest sign of cohesion, they attribute harmful intent). This would be a QUANTITATIVE difference, right?

5. Lines 285-286: The conclusion drawn here is “the scaling of the response to changes in social threat is similar in all individuals” (high vs. low paranoia individuals). Please discuss whether the study had sufficient power to observe this. As the authors probably know, power for such interaction effects (or moderation) is complicated and strongly dependent on the population differences in the slopes (e.g., Stone-Romero et al., 1994, Journal of Management, 20, 167-178). Please consider this issue and appropriately qualify your conclusions if needed. For example, maybe the data support the conclusion that there are no large differences in scaling of responses. In my view, that would be a fine conclusion.

6. Related to the point above: What was the distribution of individual differences in paranoia (GPTS scores)? Does the sample distribution or the small number of people with high scores reduce the power to find that highly paranoid individuals respond differently?

Reviewer 3 ·

Basic reporting

This is a well-written article, that does present an interesting piece of research that is written in professional English throughout.

There is (on the most part) sufficient literature/context provided on the background – however, as the focus is linked to conspiracy theory beliefs, I would like to draw the author (s) attention to a relevant paper by Imhoff and Lamberty (2018, https://doi.org/10.1002/ejsp.2494). In this paper, they propose that paranoia and belief in conspiracy theories are distinct (albeit correlated) constructs. The author (s) may wish to visit this paper, as I feel it would be useful to include this within their rationale and may also help support their conclusions (where it has been shown that paranoia did not interact).

In addition, the paper finds that harmful intentions were attributed to a cohesive group (vs non-cohesive). This reminds me of another paper that may be relevant to the author (s). Grzesiak-Feldman and Suszek (2008 https://doi.org/10.2466/pr0.102.3.755-758) simply found that conspiracy theories are seen to be more plausible if they are a tight formed group. I believe your research supports this, where you have extended this by showcasing the effects in a live interaction.

The article is structured well, where the Tables and Figures are clear. There is also clear raw data linked within the article.

The hypotheses are tested within the Results, and it is clear to see the links between each hypothesis.

Experimental design

The research is original, with a well-defined research question. There is no research to date that has examined conspiracy thinking in live interaction, so this offers novelty to the research.

The research is based on a high-powered sample, that was pre-registered. This brings in a strong sense of rigor to the research.

In the design, it would have been good if the author (s) included that they are also examining paranoia as part of their analyses. I do think this should feature in the design section.

Validity of the findings

The data is based on a high-powered sample, so I am confident that the findings are robust and statistically sound.

Conclusions are strong, where the limitations are discussed. I wonder if it could be considered a limitation that belief in conspiracy theories (or mindset) was not measured? I would predict that someone who is more likely to endorse conspiracy theories, may attribute a higher degree of harmful intent. We know from Imhoff and Lamberty that conspiracy theories and paranoia are different so I would predict that the conspiracy theory measure would have interacted.

I also wondered about a power imbalance – where from the literature, those who are feeling powerless, are more likely to believe in conspiracy theories. With the set-up of the game (2 vs 1), could this have, as an outcome of the design, bred feelings of powerlessness, leading to the harmful intent?

Additional comments

This is an interesting paper, where the novelty of examining a live interaction, within a high-powered sample sparked by interest. The paper is well-written, where some of the limitations were already addressed within the Discussion itself. I have included a couple of points above that the author (s) may wish to consider.

---

## Round 0.2 · accepted · Accept

Thank you for the revised manuscript and response letter. I am pleased to inform you that your manuscript " Paranoia and conspiracy: group cohesion increases harmful intent attribution in the Trust Game " has been accepted for publication.

·

Basic reporting

The writing is clear and the revised manuscript is very satisfactory.

Experimental design

The revisions are clear and sufficient.

Validity of the findings

Again, revisions are satisfactory.

Additional comments

Thank you for carefully addressing all the comments brought up by the reviewers.

Reviewer 2 ·

Basic reporting

no comment

Experimental design

no comment

Validity of the findings

no comment

Reviewer 3 ·

Basic reporting

No further comments.

Experimental design

No further comments.

Validity of the findings

No further comments.

Additional comments

Thank you for addressing the points of clarification that I raised in my review. I am happy with all the responses and congratulate you on an interesting piece of research.